# DNA origami cryptography for secure communication

Yinan Zhang[1,2,8], Fei Wang[1,8], Jie Chao [3], Mo Xie[2], Huajie Liu[4]*, Muchen Pan [2], Enzo Kopperger [5], Xiaoguo Liu [1], Qian Li[1], Jiye Shi [2], Lihua Wang[2,6], Jun Hu[2,7], Lianhui Wang [3], Friedrich C. Simmel [5] & Chunhai Fan [1]*

Biomolecular cryptography exploiting specific biomolecular interactions for data encryption represents a unique approach for information security. However, constructing protocols based on biomolecular reactions to guarantee confidentiality, integrity and availability (CIA) of information remains a challenge. Here we develop DNA origami cryptography (DOC) that exploits folding of a M13 viral scaffold into nanometer-scale self-assembled braille-like patterns for secure communication, which can create a key with a size of over 700 bits. The intrinsic nanoscale addressability of DNA origami additionally allows for protein binding-based steganography, which further protects message confidentiality in DOC. The integrity of a transmitted message can be ensured by establishing specific linkages between several DNA origamis carrying parts of the message. The versatility of DOC is further demonstrated by transmitting various data formats including text, musical notes and images, supporting its great potential for meeting the rapidly increasing CIA demands of next-generation cryptography.

[1] School of Chemistry and Chemical Engineering, and Institute of Molecular Medicine, Renji Hospital, School of Medicine, Shanghai Jiao Tong University, Shanghai 200240, China. [2] Division of Physical Biology, CAS Key Laboratory of Interfacial Physics and Technology, Shanghai Institute of Applied Physics, Chinese Academy of Sciences, Shanghai 201800, China. [3] Key Laboratory for Organic Electronics & Information Displays (KLOEID), Institute of Advanced Materials (IAM) and School of Materials Science and Engineering, Nanjing University of Posts & Telecommunications, 9 Wenyuan Road, Nanjing 210046, China. [4] School of Chemical Science and Engineering, Shanghai Research Institute for Intelligent Autonomous Systems, Key Laboratory of Advanced Civil Engineering Materials of Ministry of Education, Tongji University, Shanghai 200092, China. [5] Physics of Synthetic Biological Systems (E14), Physics Department, Technische Universität München, Am Coulombwall 4a, 85748 Garching, Germany. [6] Shanghai Key Laboratory of Green Chemistry and Chemical Processes, School of Chemistry and Molecular Engineering, East China Normal University, 500 Dongchuan Road, Shanghai 200241, China. [7] Shanghai Synchrotron Radiation Facility, Zhangjiang Laboratory, Shanghai Advanced Research Institute, Chinese Academy of Sciences, Shanghai 201210, China. [8] The authors contributed equally: Yinan Zhang, Fei Wang *email: liuhuajie@tongji.edu.cn; fanchunhai@sjtu.edu.cn

Information security[1]—confidentiality, integrity. and availability (the "CIA triad") of information—plays a pivotal role in modern society. In order to meet this demand, sophisticated cryptography schemes[2] relying on hard computational problems have been established for secure communication[3–5]. However, current cryptography protocols are facing severe challenges: the tremendous and ongoing progress of electronic computers[6] will soon allow to crack currently used cryptography protocols within acceptable time by brute-force attacks[7], while the emergence of novel quantum computers[8] will allow to crack keys based on prime factorization via Shor's algorithm[9]. Next-generation cryptography circumventing these threats, therefore, has received extensive attention. In particular, quantum cryptography methods exploiting the quantum mechanical uncertainty principle hold great promise for assuring message confidentiality[10]. However, whether CIA of information can be comprehensively achieved via quantum communication remains unclear.

Biomolecular cryptography that utilizes highly specific, thermodynamically controlled biomolecular interactions instead of computational schemes for encryption has been previously proposed as an alternative[11–13]. For instance, a biomolecular keypad lock that can authorize password entries was developed based on the sequential recognition of input substrates of specific biocatalysts[14]. Similarly, proteins[15], aptamers[16], bacteria[17], and DNA-based biocomputing[18] have been exploited to protect messages for secure communication. However, in previous studies, information security relied on fixed biomolecular reaction schemes, whose security would have been compromised as soon as the adversary uncovered the "trick". In 1999, Clelland et al. developed a DNA-based steganography scheme to hide secret messages[19], opening a new era of DNA cryptography that involved information-rich biomolecules for the creation of data encryption keys to ensure message confidentiality[20–23]. Nevertheless, these DNA-based strategies generally exploit sequence information only, whereas they largely ignore the structural potential of DNA.

DNA origami[24–27] is a technique for biomolecular self-assembly that generates DNA-based nanostructures through folding of a long "scaffold" strand with the help of hundreds of short "staple" strands. Its intrinsic nanoscale addressability allows the precise organization of molecules and nano-objects into complex patterns[28–34]. Here, we utilize the technique for "DNA origami cryptography" (DOC), which implements braille-like nano-patterns for robust secure communication largely meeting the CIA criteria by providing protection on confidentiality, integrity and access control. In DOC, we encrypt messages into sequential spot patterns that are implemented physically by a combination of scaffold strands each carrying a set of message-specific biotinylated strands. The message is decrypted by folding the scaffold strand into a DNA origami structure using a specific set of staple strands. The procedure is associated with a huge design space for the keys, considerably surpassing the current limitations of encryption protocols based on computational problems such as factorization. For example, folding a M13mp18 scaffold (7249 nt) corresponds to a theoretical key size of over 700 bits (see below), while AES[5] uses no more than 256 bits. The confidentiality of DOC can be further enhanced by combining it with steganography and pattern encryption enabled by DNA origami. Furthermore, message integrity can be ensured by introducing specific linkages between DNA origamis carrying parts of the message. This can also be used to realize differential access to the message—receivers (or interceptors) will retrieve different messages depending on their linker strands. Finally, we demonstrate that by reengineering the spot patterns this method is versatile in transmitting messages of different lengths and in various formats, including but not limited to text, musical notes and images.

## Results

**DOC for message confidentiality**. The workflow of confidential communication between the sender and receiver—Alice and Bob—with DOC is displayed in Fig. 1a. The whole process is composed of three layers—encryption of the message into a dot pattern as the outer layer, followed by a steganographic intermediate layer, and finally DNA origami encryption (DOE) as the innermost layer, represented by three nested channels colored in gray, green and pale green, respectively. The pattern encryption step translates the original message into a sequential pattern in order to accommodate it to the DOE scheme. As shown in Fig. 1b, Alice initially encoded the plaintext message "HEY" (Supplementary Fig. 1) letter by letter into binary numbers, followed by encryption of the numbers for each letter (in navy) and their respective positions in the message (in teal) into a braille-like spot pattern cipher. Each spot in the pattern represents a distinct digit of the binary numbers encoding the letters or their positions. The key is the permutation of the spots to represent the information (Supplementary Fig. 2). A DNA origami folding scheme was then used for the next encryption step. To this end, a custom DNA scaffold sequence was routed through a defined geometry covering the spot pattern. Importantly, the origami structure was not physically folded with DNA staple strands at this stage. Instead, a set of biotinylated message strands ("M-strands") were hybridized to the scaffold strand. For a structurally symmetric DNA origami, an additional M-strand was introduced as a marker (referred to as "MARKER") to facilitate unique identification of the pattern downstream. In this way, the original spot patterns were encrypted into a combination of scaffold strands carrying M-strands. The corresponding key is the specific folding of the scaffold with a defined length, sequence and folding shape. The biotinylated positions are invisible to potential adversaries, introducing additional protection by steganography. Each M-strand contained a three-thymine spacer close to the biotin and a segment of 40–48 nucleotides perfectly matched to the scaffold, which ensured the occurrence of biotin at the desired spot site. Due to their length, M-strands are not displaced from the scaffold by the shorter origami staple strands, when the origami structure is physically folded in a thermal annealing process from 57 °C to room temperature (which is sufficient for single-layer DNA origami folding as shown in Supplementary Fig. 3) in $1 \times$ TAE buffer containing 12.5 mM $Mg^{2+}$ (Supplementary Fig. 4). After the removal of unbound M-strands, scaffold strands carrying different M-strands were collected and delivered to Bob in a test tube or adsorbed on paper (Supplementary Fig. 5).

In contrast to Alice, Bob then physically folded the DNA origami structures with the corresponding staple strands to reveal the biotin patterns (Fig. 1c). Subsequently, streptavidin was added to the origami to make the biotin patterns recognizable under an atomic force microscope (AFM). Finally, Bob decrypted the streptavidin patterns one by one to obtain the plaintext message "HEY" based on the defined array for pattern encryption. We note that the steganography strategy can be employed also by other means than the biotin-streptavidin interaction. As shown in Fig. 1d, alternatively fluorescently labeled M-strands were used to define the DNA pattern (Supplementary Fig. 6), which could be revealed via stochastic optical reconstruction microscopy (STORM). AFM images of braille-like streptavidin patterns for the characters A to Z are shown in Fig. 1e.

**Text communication**. Secure communication of an eight-letter text is demonstrated in Fig. 2. At first, the message "19120623", the birthday of Alan Turing, was encrypted into eight sequential spot patterns (Fig. 2a). As an example, the letter "9" and its position "2nd" were separately converted to binary numbers and

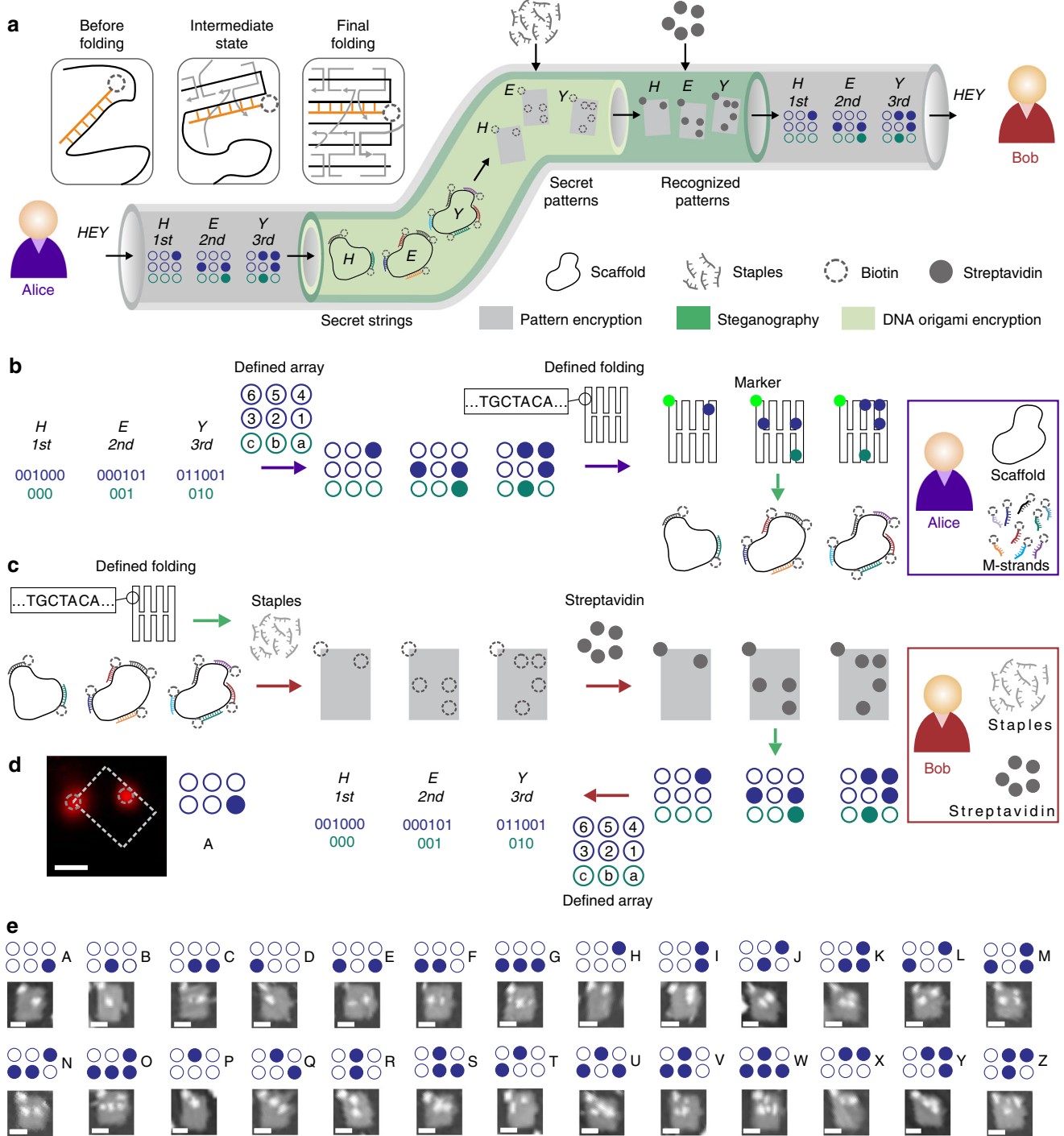

**Fig. 1** Security protocol of DOC for message confidentiality. **a** The whole process is composed of three layers—encryption of the message into a spot pattern as the outer layer, followed by a steganographic intermediate layer, and finally DNA origami encryption (DOE) as the innermost layer, represented by three nested channels colored in gray, green and pale green, respectively. **b** Encryption of the message by Alice. Alice holds the DNA scaffold and can generate the M-strands. At first, Alice encoded the plaintext message "HEY" letter by letter into binary numbers, and then encrypted the numbers for each letter (in navy) and their respective positions in the message (in teal) into a braille-like spot pattern. Afterward, Alice encrypted the patterns into a combination of scaffold strands carrying several M-strands, according to a defined DNA origami folding scheme. **c** Decryption of the message by Bob. Bob holds streptavidin and can generate the staples. With the staples Bob was able to fold the DNA origami, revealing biotinylated patterns on the M-strands. Subsequently, Bob added streptavidin to make the patterns recognizable under the AFM. Finally, the plaintext message was decrypted letter by letter into binary numbers and decoded. **d** The fluorescent pattern under the STORM. Scale bar: 50 nm. **e** Braille-like streptavidin patterns under the AFM. Scale bar: 50 nm.

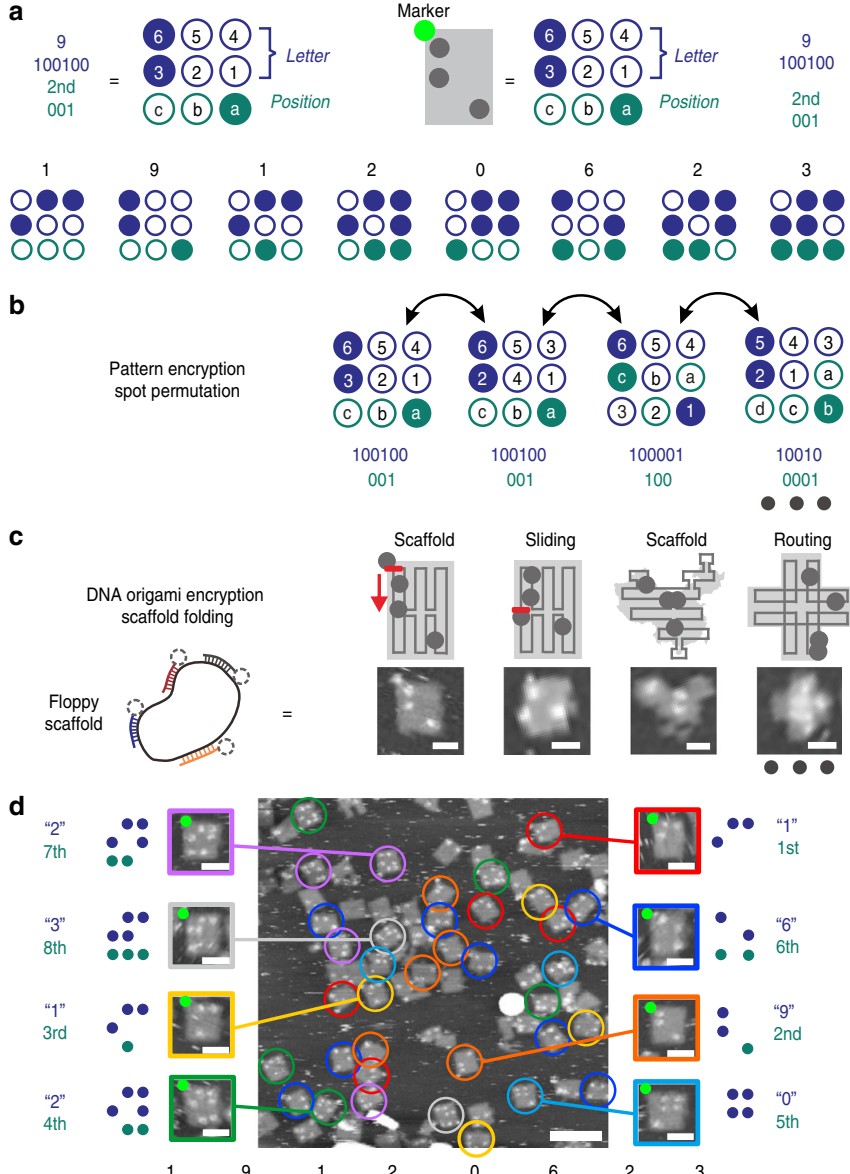

**Fig. 2** An eight-letter text communication. **a** Spot patterns for encryption of the text. Generation of key space in the pattern encryption (**b**) and DNA origami encryption (**c**). **d** AFM images of streptavidin patterns conveying the encrypted message. Major streptavidin patterns with each position number were circled in different colors. Scale bar: 200 nm (inset: 50 nm).

represented with a spot pattern. The key space for pattern encryption derives from the number of possible permutations of the spots representing the binary digits of the letters or their positions (Fig. 2b). Correspondingly, the key size is given by

$$K_{PE} = \log_2 \sum_{i=1}^{m} P_i^m P_{m-i}^{m-i} = \log_2(mP_m^m), \quad (1)$$

where $m$ is the number of involved spots and $P_i^m$ is the number of $i$-permutations of $m$. In this study, we adopted a value of 9 for $m$, which resulted in a key size of pattern encryption at about 22 bits. More importantly, pattern encryption provided an interface between message and DNA origami encryption, which plays a key role in our scheme for secure communication.

DOE is based on the unpredictability of the sequence, length, and folding of the scaffold strand. A brute-force attack on the scaffold or staple strands is not practically feasible (see Supplementary Discussion for details). Suppose the presence of a

powerful adversary ("Mallory") who can somehow intercept the scaffold strand transmitted from Alice to Bob. In practice, the chance to replace the DNA media by a counterfeit during delivery is little. Laborious sequencing is required to find out the length and sequence of the scaffold strand (Supplementary Fig. 7). After that, Mallory would have to crack the specific routing and sliding of the scaffold in the DNA origami using an exhaustive method (Supplementary Fig. 8). Any variation on either of the factors would result in a detectable variation of the pattern (Fig. 2c, Supplementary Figs. 9–12). A simplified model predicts that the key size could reach up to 702 bits for a 7249-nucleotide M13mp18 scaffold (Supplementary Equation 5), which is a significant advance compared to AES which works with a key size of maximally 256 bits. For longer scaffolds such as p7560 and p8064, the theoretical key sizes are 732 and 780 bits, respectively.

This key size is based on the theoretical number of possible biotin patterns on a DNA origami sheet, which in practice could be slightly reduced considering the finite size of streptavidin for

pattern decryption (Supplementary Fig. 13), imaging resolution, and due to other experimental restrictions. To overcome the limitation of AFM imaging and the structural variation, we optimized the system in several aspects. First, the neighboring streptavidin spots were placed far apart from one another to make them discernible by AFM imaging. Second, only a short spacer ($T_3$) was incorporated in the biotinylated M-strand. The rigid short spacer prevented the fluctuating of streptavidin spots. Third, near-minimum-force was exerted on the sample during AFM imaging. We thus believe that DNA origami encryption is capable of providing stronger protection on the confidentiality of message than AES.

Figure 2d presents the AFM images of streptavidin patterns on the rectangular DNA origami (Supplementary Fig. 14). Only well-folded structures carrying the MARKER streptavidin were considered in the decryption. Every pattern with MARKER was recognized according to its position number. For example, the patterns with MARKER and all the three position spots occupied were taken into account for decryption of the last letter. The majority of patterns considered for each position were accepted as the encrypted letter to be decrypted. Statistics on thirty AFM images show patterns at the eight positions, each taking an overall percentage from 57.6 to 79.4% (Supplementary Figs. 15, 16). In the rejected fraction of patterns, some of the streptavidin spots were missing, which we attribute to three major factors: the incompleteness of biotin-streptavidin conjugation (~95%)[15], undesired dissociation of some of the M-strands from the scaffold, and a possible mechanical removal of streptavidin by the AFM tip (see Supplementary Discussion on "Bit Error Probability"). Decrypting the patterns (circled in different colors for each position in Fig. 2d) resulted in the plaintext message "19120623" conforming to the original one, implying that DOE maintained the message during the communication. Two blind tests in which the receiver was not informed of the content of message previously were performed. In order to prevent the receiver from being misled by the minority of wrong patterns, we set the lower limit for the number of patterns required for each position to 20. In both the two blind tests, the receiver successfully decrypted the right message (Supplementary Figs. 17–20), further confirming the feasibility of DOE.

**Integrity and differentiated access.** To further enhance the security level of DOC, we introduced specific recognition interactions between different DNA origami structures to maintain the integrity of the messages and achieve differential access—i.e., different entities will have different access to the encoded information. As an example, the intelligence of Operation Overlord carried out by the Allied Forces during World War II is represented with a spot pattern as shown in Fig. 3a. A map of the coastline of England or France, respectively, is depicted on different DNA origamis with dumbbell-shaped bulge loops on selected staple strands (Fig. 3b and Supplementary Fig. 21). The corresponding origami structures are referred to as E-tiles and F-tiles, respectively. Locations at the coast are denoted by red spots in the map, while dates encoded as binary numbers are represented by green spots flanking the E- and F-tiles (left: month, right: day). Purple spots are used to generate a hash value of the message for maintaining its integrity, where each spot is indexed to a digit in a binary number denoting an alphabetical letter (Supplementary Fig. 22). The E–F tile dimer with a hash value segment "E–F" carries the expected location and date of the launch of Operation Overlord. There are four pairs of 8-nt sticky ends at the bottom of the E-tiles and the top of the F-tiles, which facilitate the dimerization of E- and F-tiles upon the addition of linker strands. Differential access to the message is achieved by

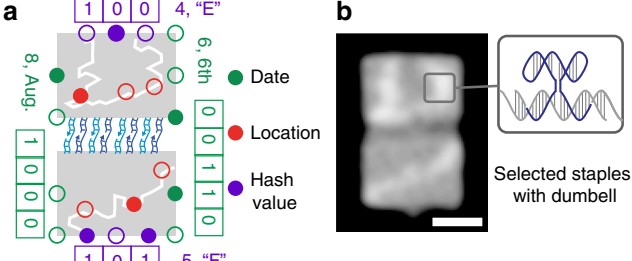

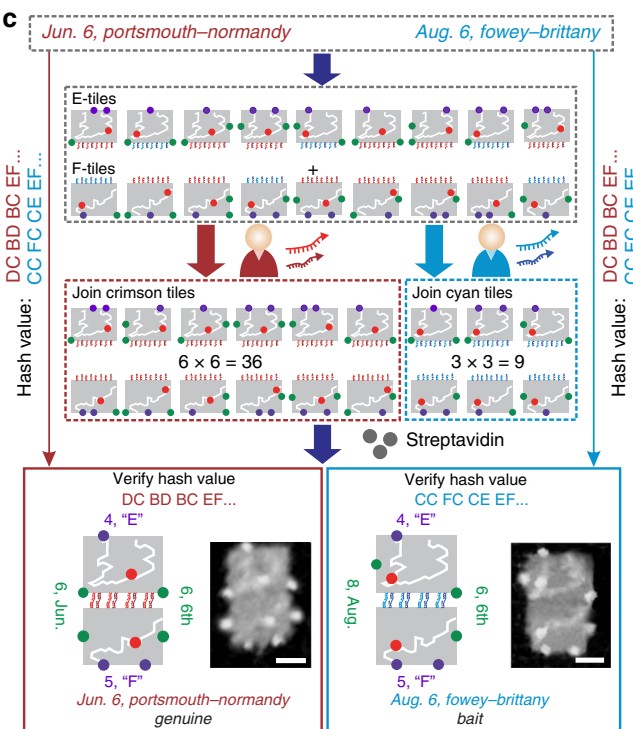

**Fig. 3** Maintaining the integrity of message and achieving differentiated access. **a** Presentation of the message with a spot pattern. Hash value and password protection were introduced. **b** The England and France maps depicted on E- and F-tiles are fabricated with DNA dumbbell-shaped bulge loops on selected staples, respectively. **c** Transmitting process of the message. Hash value is used to verify the integrity of message. Bob has access to the genuine message while Mallory is led to the bait one. Scale bar: 50 nm.

introducing two sets of sticky ends to different combinations of DNA origamis and distributing different sets of linker strands to Bob and Mallory, respectively (Fig. 3c). The linker strands act as a password to provide Bob and Mallory with different access to the message, which ensures Bob has access to the genuine message, while Mallory is deceived with a bait message.

First, the concatenated message "Jun. 6, Portsmouth-Normandy; Aug. 6, Fowey-Brittany" was encrypted into nine patterns on E- and F-tiles, respectively, with each tile containing a hash value segment. The hash values were generated from dimerization of E- and F-tiles (see Supplementary Discussion on collision resistance of the hash algorithm). Meanwhile, the original hash values corresponding to the portions of the message destined for Bob and Mallory, respectively, were both distributed to them for verification. Both of the hash values are defined as correct (Supplementary Fig. 23). Assume Mallory has intercepted all the keys for encryption, he would obtain the same combination of biotin patterns on mixtures of E- and F-tiles as Bob (Fig. 3c). However, Bob's linker strands selected the crimson group of E–F

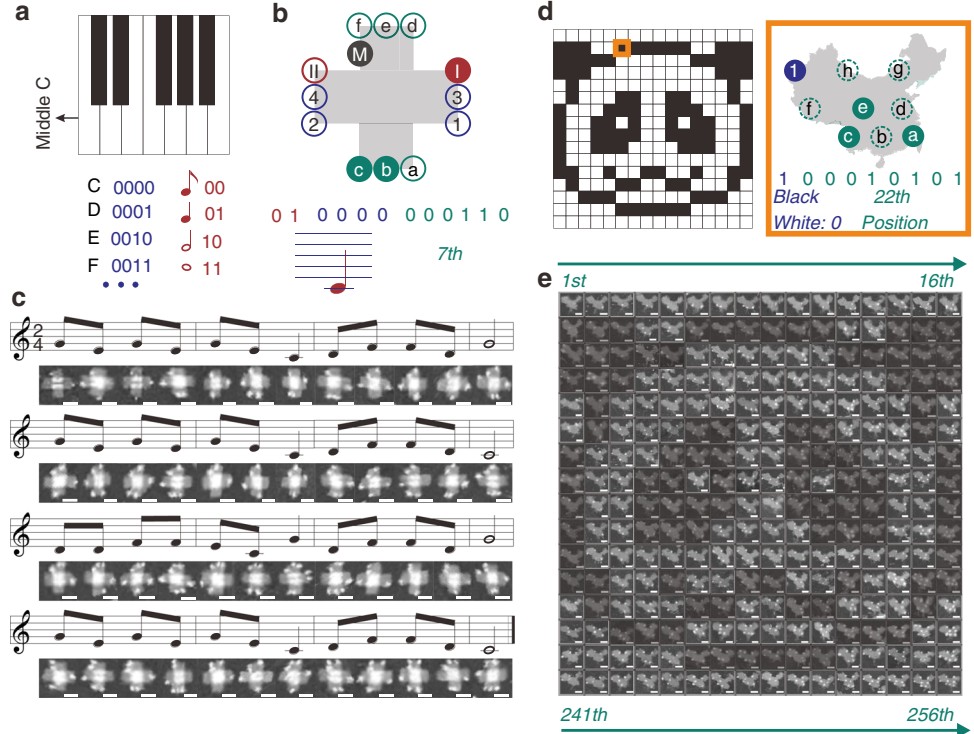

**Fig. 4** Transmitting music and image with DOC. **a** The custom keypad indexing the music to binary numbers. **b** From spot pattern to music. **c** AFM images of streptavidin patterns conveying the music. (**d**) A 256-pixel panda image. The pixel boxed in orange is presented by a spot pattern (an analogic map of China). (**e**) 256 streptavidin patterns conveying the panda. Scale bar: 50 nm.

tile dimers carrying the genuine message while Mallory's selected the cyan tiles carrying the bait message (Fig. 3c). The yields of dimerization of E- and F-tiles for Bob and Mallory were analyzed under the AFM and estimated to be 69.8 and 71.3%, respectively (Supplementary Fig. 24).

After dimerization of the origami tiles, streptavidin was introduced to reveal the patterns (Supplementary Fig. 25). The generated hash values were verified by comparison with the original one. Streptavidin patterns with hash value segment "EF" were chosen and identified (2.8% for Bob and 11.1% for Mallory in theory). The location was read from the map and the associated date was obtained from the binary number. In total, the message "Jun. 6, Portsmouth-Normandy" was transmitted to Bob while the message available to Mallory was "Aug. 6, Fowey-Brittany" (Fig. 3c). We find that the password strength reaches 128 bits, providing sufficient protection of the message via access control (Supplementary Equation 17).

**Versatility of DOC.** We further demonstrate DOC as a versatile method for the transmission of other data formats such as musical notes and images. Figure 4a shows a schematic representation of the one-line octave of a piano keyboard and rules for the conversion of the individual pitches and note values into binary numbers (Supplementary Fig. 26). A twelve-spot pattern on a cross-shaped DNA origami[35] was used to encrypt the numbers (Fig. 4b and Supplementary Fig. 27). The two spots in crimson cover four note values from whole to the eighth. Pitches are represented by the four flanking spots depicted in navy. Dotted, double and triple dotted notes are included here, as well. The remaining six spots in teal are used to represent the position of a note in a music score. An extra spot is used as MARKER. In the example shown in Fig. 4b, the spot pattern is decrypted as a quarter note at "C" at the 7[th] position of the music score. In this

way, the children's song "House Painter" is represented with streptavidin patterns (Fig. 4c).

In another example, a 256-pixel image of a panda image was transmitted with DOC (Fig. 4d). Individual pixels were converted into binary numbers, encoding "black" and "white" to "1" and "0", respectively. The pixels of the image were numbered from left to right row by row, with the position number increasing from 0 to 255. A DNA origami structure shaped as an analog map of China[36, 37] carrying a nine-spot pattern was used to transmit the image. The MARKER was omitted due to the intrinsic asymmetry of this origami structure (Supplementary Fig. 28). The upper left spot in navy is used to represent the color while the other eight spots in teal represent the position of the pixel ($2^8 = 256$ pixels in total). Hence, the pattern in orange box in Fig. 4d is decrypted as "the 22nd pixel is black". Figure 4e displays all the 256 streptavidin patterns constituting the panda image.

In order to collect the streptavidin patterns, multiple AFM scans of the samples were required. Due to the diversity of the involved patterns, errors in identifying patterns for each position occurred in initial scans (Supplementary Figs. 29, 30), which was corrected when larger numbers of patterns were accumulated. For the 48-note music and 256-pixel image, 25 and 70 scans were performed with 1.5-nM sample in a size of 2 μm × 2 μm to collect enough patterns for a correct decryption of the music score and the image, respectively (Supplementary Figs. 31, 32).

## Discussion
Our work demonstrates a cryptography method that introduces DNA origami to provide multi-level protection of messages for secure communication. Messages were translated into secret braille-like patterns in order to facilitate DOE with a large key size (using a 7249-nt M13mp18 scaffold corresponds to a theoretical key size of over 700 bits). Protein binding-based steganography additionally assured the confidentiality of message. Further, by

exploiting specific hybridization-based recognition between different DNA origamis, maintaining the integrity of a message and differentiated access was achieved. Different types of messages including text, musical notes, and images were transmitted with DOC, manifesting it as a universal method.

In order to develop our approach into a practical data encryption technique, other molecular markers instead of streptavidin could be used to encode the nano-patterns conveying a message. The next steps would be to attempt detection of the spot patterns at a higher resolution and to also employ 3D characterization methods. With a higher resolution, a larger number of spots can be embedded in the patterns, which would improve pattern encryption and, more importantly, increase the information storage capacity of the structures. Using 3D characterization methods would allow to also use 3D routings of the DNA scaffold, which could increase the key size by orders of magnitude.

The decryption time in the present work normally took 1–2 h for each pattern, including sample processing, AFM scanning, human-based identification, and readout. Although the decryption time is long as compared to that using electronic computers, the DOC encryption provides a biomolecular solution to comprehensive and strong protection excelling the widely used AES system. We also note that advances in high-speed AFM, automated sample processing and fully-computerized data and image analysis would greatly improve the decryption speed.

Different from previous chemical or biological encryption methods[20, 38, 39], the DOC method uses information-based DNA self-assembly to create physical puzzles, resulting in extraordinarily strong all-around protection of a secret message. Although the hypothetically huge key space of DOC cannot yet be exploited due to the intrinsic limitations of the characterization methods (e.g., AFM or STORM), DOC provides a biomolecular solution to comprehensive and strong protection, holding the potential for meeting the high CIA demands for next-generation information security.

## Methods

**Materials**. Biotinylated DNA strands purified by HPLC were purchased from Sangon Biotechnology. Alexa 647-labeled DNA strands purified by HPLC were purchased from Invitrogen. Unmodified DNA strands purified by PAGE were purchased form JieLi Biology. Streptavidin was purchased from Sigma-Aldrich. M13mp18 scaffold strands were purchased from New England Biolabs. Fluo-Sphere[TM] carboxylate-modified microspheres were purchased from Thermo Fisher. All other chemicals were purchased from Sinopharm. Water was purified with a Millipore Milli-Q Integral water purification system (resistivity = 18.2 MΩ·cm).

**Choosing an appropriate scaffold**. A geometry that covered the spot pattern conveying the message was firstly defined. Secondly, the scaffold length fitting the defined geometry in a raster-filling method was determined. A short scaffold (below 150 nt) can be synthesized chemically. Fabrication of longer scaffold can be achieved from a natural plasmid template. The scaffold used here, M13mp18, was commercially achievable.

**Generating sequences of M- and staple strands**. Sequences of M- and staple strands depended on the scaffold of DNA origami. Previously, the scaffold was folded back and forth to fill the defined geometry covering the spot pattern, revealing the correspondence between M-strands and spots. Therefore, the sequence of M-strands at individual spots were generated from the scaffold. Sequences of staple strands can be generated based on the scaffold folding with aid of professional software such as caDNAno. However, a DNA origami design not restricted to classical models needs to be finished manually. It should be noted that the M-strands hybridized with the scaffold are forbidden to hinder scaffold crossovers in DNA origami. Nevertheless, some staple crossovers are inevitably hindered by the spanning M-strands. Elaborate adjustment was performed to arrange M-strands on the scaffold at minimum sacrifice of staple crossovers. No evident damage caused by loss of local staple crossovers to DNA origami was observed in the experiment.

**Binding M-strands to DNA scaffold**. 200 nM M-strands were mixed with 20 nM scaffold strands in $1 \times$ TAE buffer (40 mM Tris, 20 mM acetic acid, 2 mM EDTA, pH 8.0) with 12.5 mM $Mg^{2+}$. Excess M-strands facilitated a complete hybridization with the scaffold. A rapid anneal from 85 °C to 4 °C was then performed. Afterward, unbound M-strands were removed by centrifuge filters. The molecular weight cut-off (MWCO) of centrifuge filters depends on the length of DNA scaffold. 100 kDa is ideal for M13mp18 scaffold.

**Delivery of DNA scaffold**. Scaffold strands carrying different M-strands can be collected in tube and directly delivered to Bob. Alternatively, the collected mixture can be dropped onto a paper for delivery. The dropped spot on the paper was cut and soaked in $1 \times$ TAE buffer with 12.5 mM $Mg^{2+}$ for 30 min. After that, the remnant was squeezed and the supernatant was collected. Fresh buffer was then added to rinse the remnant. After three times of rinsing, the collected supernatant was concentrated to a final concentration of 20 nM.

**DNA origami folding**. Staple strands were preheated to 95 °C for 3 min and cooled to room temperature slowly. Scaffold strands carrying M-strands were then mixed with the staples at a molar ratio of 1:10 in $1 \times$ TAE buffer with 12.5 mM $Mg^{2+}$. The final concentration of scaffold strands was maintained at 2 nM. Afterward, the mixture was heated at 57 °C for 3 min and then annealed to 27 °C at a rate of −5 °C min$^{-1}$. Folded DNA origami was then purified with 100 kDa (MWCO) centrifuge filters three times to remove excess M- and staple strands.

**Joining E- and F- tiles**. Linker strands were mixed with E- and F-tiles at ten times the concentration of the sticky ends. The mixture was then annealed slowly from 45 to 25 °C in three cycles and finally held at 25 °C.

**Adding streptavidin to recognize biotin patterns**. Streptavidin was added to recognize the biotin patterns on DNA origami at a molar ratio of 10:1 to the biotin on DNA origami. After a 2-h incubation at room temperature, the patterns were characterized under the AFM.

**AFM imaging**. A droplet (~2 μL) was deposited on freshly cleaved mica surface and left to absorb for 3 min. After that 40 μL of $1 \times$ TAE buffer containing 12.5 mM $Mg^{2+}$ was added to the liquid cell and a NP-S (Bruker, Inc.) tip was used to scan the sample in a PeakForce-tapping mode on a Multimode VIII AFM (Bruker, Inc.). A minimum force was maintained in imaging to prevent scratching of streptavidin by the tip which could led to false negative results. DNA origami showed a high tendency to aggregate in $1 \times$ TAE buffer containing 12.5 mM $Mg^{2+}$. Removal of staples strands binding at the edge of DNA origami from the staple library alleviated the aggregation. Nevertheless, AFM characterization should be undertaken soon after the addition of streptavidin to DNA origami. Undistinguishable patterns on aggregation of DNA origamis were excluded from statistics.

**STORM imaging**. Alexa 647-labeled DNA strands were added at ten times the concentration of the anchors on DNA origami. After overnight incubation at 25 °C, free strands were removed with 100 kDa (MWCO) centrifuge filters. The DNA origami was dropped on a glass dish at a concentration of ~100 pM. Before the deposition, the glass dish was treated with negative glow discharge. FluoSphere[TM] carboxylate-modified microspheres were used as the drift marker. Imaging was performed with inclined illumination at an excitation intensity of 200 W cm$^{-2}$ at 488 nm and 647 nm. Images were reconstructed from more than 30,000 frames at an interval of 20 ms. ImageJ was used for image processing with Gaussian fitting algorithms.

## Data availability

The data presented in this paper are available from the corresponding authors upon reasonable request. The source data underlying Supplementary Figs. 4, 5, 16, 18, 20, 31, and 32 are provided as a Source Data file. Any other relevant data are available from the authors upon reasonable request.

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

## Acknowledgements

We thank Nadrian Seeman and Hendrik Dietz for discussion and kind suggestion. The work was supported by National Key R&D Program of China (2018YFA0902600) and National Natural Science Foundation (21834007, 21722310, 21675167, 21603262), K.C. Wong Foundation at Shanghai Jiao Tong University and Innovative research team of high-level local universities in Shanghai. E.K. and F.C.S. acknowledge support by the Deutsche Forschungsgemeinschaft through the SFB 1032/TP A2.

## Author contributions

C.F. supervised the research. Y.Z., H.L. and C.F. conceived the concept. Y.Z., H.L., J.C., M.X. and F.W. performed the experiments. M.P., E.K., X.L., Q.L., J.S., Lih.Wang., J.H., Lia.Wang assisted AFM and TEM imaging and discussed on the data. Y.Z., H.L., F.C.S., and C.F. wrote the paper. All authors provided input during manuscript writing.

## Competing interests

The authors declare no competing interests.
