## [Peer Review File · Nature Communications]

Reviewers' Comments:

Reviewer #1:

Remarks to the Author:

The authors present an interesting approach for cryptography realized with DNA origami nanotechnology. It is a nice combination of programmed patterning on structured DNA assemblies and several data encryption approaches. Overall, the paper is written clearly and the main features of the proposed cryptography can be easily understood from illustrative examples from the text to musical notes and images. I think the contents in this manuscript would be well received by the broad readership of Nature Communications. Nevertheless, there are few points that need to be properly addressed by the authors.

1. In Supplementary Fig. 15, it would be essential to show the micrographs of rejected patterns and the percentage of the second (and third) majority pattern(s) as well. More importantly, it needs to be clarified whether the percentages of major patterns were evaluated only with well-folded structures or mis-folded structures were counted as rejected. It is confusing whether the rejected fraction of patterns is purely due to missing streptavidin spots in the properly folded structures or includes the portion of poorly folded structures as well.
2. One concern I have is that the majority pattern approach in decryption might be limited when the message has the same letter at different positions. For example, if the message were "11121623" (the same letter 1 at different positions of 1, 2, 3, 5), the structures with missing spot markers at the positions 2, 3, and 5 would be counted as the successful letter at the position 1 contaminating the statistics. How will Supplementary Fig. 15 look like if the message is, for example, "11112222" or "12121212", etc.?
3. Another concern is how accurately we can distinguish the location of markers on the origami. Due to the limitation of AFM imaging (resolution, noise, structure distortion, etc.) as well as the structural variation, marker points in some structures might be hardly discernible. For example, in Fig. 1e, AFM images of A, B, D, P (or C, E, F) can be looked similar to one another. How can we avoid this issue?
4. In Fig. 2d, are magnified sample images corresponding to the images connected by lines? They don't look exactly the same... some images seem mirrored ones, etc.
5. In Fig. 3, was the majority approach used as before? What are the percentage of it for dimers with EF hash vaule?

Reviewer #2:

Remarks to the Author:

Fan and co-workers present a scheme of DNA origami cryptography (DOC) for data encryption. Three main aspects, confidentiality, integrity and generality of such a framework are demonstrated. The data communication is implemented by AFM and STORM imaging. A simple message is coded and visualized as braille-like streptavidin patterns or STORM patterns based on fluorescent labeling. Information richer contents, including a music score and a 256-pixel image are also showcased. It is a novel concept and the implementation is solid and impressive.

Specific comments and suggestions

1. Key size. The authors claim that the key size can reach 8861 bits for a commonly used M13mp18 DNA scaffold of 7249-nt. Restricted by the practical challenges of AFM or STORM imaging however,

the theoretic key size may be difficult to achieve. The password strength of 128 bits shown in figure 3 is actually based on two combined origami instead of one. I suggest the authors to tune down such a claim. Moreover, since the key size would become larger with a larger origami structure, it would be nice if a larger origami from longer scaffold can be shown with stronger password strength.

2. Speed. AFM images of braille-like streptavidin patterns for the characters A to Z are shown in this article, and it is such a comprehensive set of results. I suppose it could take a long time to decrypt a typical message, such as a text message, a musical score and a pixelated image highlighted in this article. Can the authors compare the readout speed of DOC to common encryption protocols employed in electronic computers?

3. Accuracy. Based on the efficiency estimation of biotin-streptavidin binding of 57.6%-79.4%, how accurate it is to call a letter purely based on reference-free imaging results? Is it possible to make the right call when different letters are mixed together? In general, what level of labeling efficiency is necessary for a reference-free readout? What level of redundancy is necessary to ensure a reliable data communication? I strongly suggest a blind test (one who image the samples does not know what patterns to look at). If experiments were designed this way, the concept would be conveyed more convincingly.

In general, I suggest the authors to carefully describe the limitations and challenges of the method upfront to make sure that the general audience would not get misled by the bold claims.

Responses to the reviewers:

Reviewer #1

The authors present an interesting approach for cryptography realized with DNA origami nanotechnology. It is a nice combination of programmed patterning on structured DNA assemblies and several data encryption approaches. Overall, the paper is written clearly and the main features of the proposed cryptography can be easily understood from illustrative examples from the text to musical notes and images. I think the contents in this manuscript would be well received by the broad readership of Nature Communications. Nevertheless, there are few points that need to be properly addressed by the authors.

Reply: We are grateful for the reviewer's very positive comments.

Major points

1. In Supplementary Fig. 15, it would be essential to show the micrographs of rejected patterns and the percentage of the second (and third) majority pattern(s) as well. More importantly, it needs to be clarified whether the percentages of major patterns were evaluated only with well-folded structures or mis-folded structures were counted as rejected. It is confusing whether the rejected fraction of patterns is purely due to missing streptavidin spots in the properly folded structures or includes the portion of poorly folded structures as well.

Reply: We highly appreciate the constructive suggestion from this reviewer. Both the micrographs of rejected patterns and the percentage of the second (and third) majority pattern(s) have been demonstrated in the revised Supplementary Fig. 15. The percentages of major patterns were evaluated only with well-folded structures that carry the MARKER streptavidin. To clarify that, we added the description in the text (Line 156-157, Page 4). We defined the rejected fraction of patterns as the well-folded DNA origamis with missing streptavidin spots, as stated in Line 162-166, Page 4.

2. One concern I have is that the majority pattern approach in decryption might be limited when the message has the same letter at different positions. For example, if the message were "11121623" (the same letter 1 at different positions of 1, 2, 3, 5), the structures with missing spot markers at the positions 2, 3, and 5 would be counted as the successful letter at the position 1 contaminating the statistics. How will Supplementary Fig. 15 look like if the message is, for example, "11112222" or "12121212", etc.?

Reply: We agree with the reviewer that the structures with missing spot markers have a negative effect on the statistics. Therefore, on the one hand, we used a large excess of marker molecules (streptavidin) to ensure high binding efficiency to spot. On the other hand, we only counted the patterns with a MARKER streptavidin. For the tested message "19120623", it contains two "1" and two "2". From our statistic results

shown in Supplementary Fig. 15, the first major patterns in each position are identically the right patterns for decryption. In addition, according to the new blind test, the major patterns are also the right ones. Hence, we believe that for other messages, the major patterns in each position should also be the expected ones. As an example, the statistics of patterns at each position for the message “11112222” are shown below, which show good accordance with the message.

3. Another concern is how accurately we can distinguish the location of markers on the origami. Due to the limitation of AFM imaging (resolution, noise, structure distortion, etc.) as well as the structural variation, marker points in some structures might be hardly discernible. For example, in Fig. 1e, AFM images of A, B, D, P (or C, E, F) can be looked similar to one another. How can we avoid this issue?

Reply: We thank the reviewer for pointing out this important issue. To overcome the limitation of AFM imaging and the structural variation, we optimized the system in several facets. First, the neighboring streptavidin spots were placed far apart from one another to make them discernible by AFM imaging. Second, only a short spacer (T_3) was incorporated in the biotinylated M-strand. The rigid short spacer prevented the

fluctuating of streptavidin spots. Third, near-minimum-force was exerted on the sample during AFM imaging. New explanations have been added in the manuscript (Line 147-152, Page 4).

4. In Fig. 2d, are magnified sample images corresponding to the images connected by lines? They don't look exactly the same... some images seem mirrored ones, etc.

Reply: We thank the reviewer for pointing out this problem. Fig. 2 has now been revised such that the magnified sample images corresponds to the images connected by lines.

5. In Fig. 3, was the majority approach used as before? What are the percentage of it for dimers with EF hash value?

Reply: Yes, the majority approach was the same. The percentage of dimers with EF hash value is 2.8% and 11.1% in theory for Bob and Mallory, respectively. The description has been added to the text (Line 209-210, Page 5).

Reviewer #2

Fan and co-workers present a scheme of DNA origami cryptography (DOC) for data encryption. Three main aspects, confidentiality, integrity and generality of such a framework are demonstrated. The data communication is implemented by AFM and STORM imaging. A simple message is coded and visualized as braille-like streptavidin patterns or STORM patterns based on fluorescent labeling. Information richer contents, including a music score and a 256-pixel image are also showcased. It is a novel concept and the implementation is solid and impressive.

Reply: We are thankful that the reviewer appreciates our work and provides very positive comments.

Specific comments and suggestions

1. Key size. The authors claim that the key size can reach 8861 bits for a commonly used M13mp18 DNA scaffold of 7249-nt. Restricted by the practical challenges of AFM or STORM imaging however, the theoretic key size may be difficult to achieve. The password strength of 128 bits shown in figure 3 is actually based on two combined origami instead of one. I suggest the authors to tune down such a claim. Moreover, since the key size would become larger with a larger origami structure, it would be nice if a larger origami from longer scaffold can be shown with stronger password strength.

Reply: We highly appreciate the constructive suggestion from this reviewer. The key size is derived from the arbitrary 2D folding of the scaffold in a torsion-free manner. According to the suggestion, we modified the estimation method to show a more realistic value. The key size of 8861 bits has been reduced to 702 bits in the revised manuscript. We also agree with the reviewer that the key size is dependent on the length of the scaffold. In the present work, we employed M13mp18 DNA scaffold since it is the most widely used one for DNA origami. We believe that it will be able to create a larger biomolecular key size with longer scaffolds. The theoretical key size strengths of other scaffolds have been calculated (Line 142-144, Page 4).

2. Speed. AFM images of braille-like streptavidin patterns for the characters A to Z are shown in this article, and it is such a comprehensive set of results. I suppose it could take a long time to decrypt a typical message, such as a text message, a musical score and a pixelated image highlighted in this article. Can the authors compare the readout speed of DOC to common encryption protocols employed in electronic computers?

Reply: We thank the reviewer for this constructive comment. The decryption time in the present work normally takes 1-2 hours for each pattern, including sample processing, AFM scanning, human-based identification and readout. Although the decryption time is long as compared to that in electronic computers, the DOC

encryption provides a biomolecular solution to comprehensive and strong protection excelling the widely used AES system. We also note that the advances in high-speed AFM, automated sample processing and program-based data and image processing would greatly improve the decryption speed. (Line 263-269, Page 7)

3. Accuracy. Based on the efficiency estimation of biotin-streptavidin binding of 57.6%-79.4%, how accurate it is to call a letter purely based on reference-free imaging results? Is it possible to make the right call when different letters are mixed together? In general, what level of labeling efficiency is necessary for a reference-free readout? What level of redundancy is necessary to ensure a reliable data communication? I strongly suggest a blind test (one who image the samples does not know what patterns to look at). If experiments were designed this way, the concept would be conveyed more convincingly.

Reply: We highly appreciate the reviewer's suggestion for blind tests. The efficiency of bioconjugation between the biotin and streptavidin is approximately 95%, as reported by Wong et al. (Wong et al., *J. Am. Chem. Soc.* **2013**, 135 (8), 2931-4). In our experiment, we found it was enough to convey the message based on biotin-streptavidin interaction. To avoid being misled by the minor wrong streptavidin patterns, we performed multiple scans in different parts of the AFM mica to collect patterns for every position. In the 8-letter case, we took about 30 images for the statistics. In the musical notes and the panda image case, we found that 25 and 70 scans were enough for correct decryption, respectively. In the new blind test, we instead present a more practical method to inform the receiver of the redundancy necessary to ensure message correction: that is, the receiver should collect at least 20 patterns for each position. Two blind tests including a 5-letter and an 8-letter message were conducted. The results are demonstrated in Supplementary Figs. 16-19. We have added corresponding description in the main text (Line 168-172, Page 4-5).

In general, I suggest the authors to carefully describe the limitations and challenges of the method upfront to make sure that the general audience would not get misled by the bold claims.

Reply: We agree with the reviewer's suggestion. In the revised manuscript, we added the discussions on limitations and challenges .

“The decryption time in the present work normally takes 1-2 hours for each pattern, including sample processing, AFM scanning, human-based identification and readout. Although the decryption time is long as compared to that in electronic computers, the DOC encryption provides a biomolecular solution to comprehensive and strong protection excelling the widely used AES system. We also note that the advances in high-speed AFM, automated sample processing and program-based data and image processing would greatly improve the decryption speed.” (Line 263-269, Page 7)

“Although the huge key space of DOC has yet to be exploited due to the intrinsic limitations of the characterization methods (e.g. AFM and STROM) exist, DOC

provides a biomolecular solution to comprehensive and strong protection.” (Line 272-275, Page 7)

Reviewers' Comments:

Reviewer #1:

Remarks to the Author:

The revised paper properly addressed this reviewer's comments and hence is recommended for publication.

Reviewer #2:

Remarks to the Author:

The authors have addressed my concerns carefully in the revised manuscript. I recommend publication of this work in Nature Communication.

Reviewer #3:

Remarks to the Author:

I am not familiar with the biochemistry aspects thus my comments are strictly security related. In my opinion, a major revision is needed because the paper does not clarify the required elements of secure communications and instead presents a significant amount of redundant information for the show and media. Note: some mistakes must also be corrected, and after the required specifications more mistakes may turn up.

The comments are below:

- Abstract redundancy: Lines 34-36: As soon as a binary pattern can be transferred, it is obvious that any other data format can be transferred, including musical notes and images; facts that we know since Shannon.
- Lines 42-44: Wrong statement. Quantum computers will not be able to help with brute force attack. They will help with cracking the secure key exchange by solving related computationally hard math problems, such as integer factorization. Then the cyphertext can immediately be cracked without brute force.
- Lines 44-48: Information theoretically secure quantum and classical physical key exchangers like QKD, KLJN, or just sharing a USB stick containing the key, may indeed have problem with the accessibility part of CIA. However so is the scheme described in the present paper therefore this work is not helping that issue.
- Lines 85-89 and 90-109: The manuscript does not clarify the basic elements of secure communication even though it talks about "key" many times. Please exactly specify what is the cypher and what is the shared secret (key) between Alice and Bob and that how is that key shared.
- Section "Text communication": The same here. Please identify clearly which elements of the multi-step encoding process are using a shared secret (key) and which part are "built-in" features. Note: in accordance with Kerckhoffs's principle (Shannon's maxim) of secure communicators, only the spontaneously generated key is secure, the rest of the system is known by the eavesdropper.
- Section "Integrity ...": Please clarify how the hashing and the given signature will protect the integrity in the light of the Kerckhoffs's principle described above. The only thing that Eve will not

know is the secure key (to be specified).

- An essential feature is not characterized: Bit Error Probability. This can be a killer. Please clarify it.
- Section "Versality": As mentioned above, this is trivial though the non-expert media maybe interested. If the above-required clarifications occupy too much space, this section can be dropped.

Reviewers' comments:

Reviewer #1 (Remarks to the Author):

The revised paper properly addressed this reviewer's comments and hence is recommended for publication.

Reply: We are grateful for the reviewer's very positive comments.

Reviewer #2 (Remarks to the Author):

The authors have addressed my concerns carefully in the revised manuscript. I recommend publication of this work in Nature Communication.

Reply: We are grateful for the reviewer's very positive comments.

Reviewer #3 (Remarks to the Author):

I am not familiar with the biochemistry aspects thus my comments are strictly security related. In my opinion, a major revision is needed because the paper does not clarify the required elements of secure communications and instead presents a significant amount of redundant information for the show and media. Note: some mistakes must also be corrected, and after the required specifications more mistakes may turn up.

The comments are below:

- Abstract redundancy: Lines 34-36: As soon as a binary pattern can be transferred, it is obvious that any other data format can be transferred, including musical notes and images; facts that we know since Shannon.

Reply: We agree with the reviewer that any format of data can be presented with binary bits. Also, the text, musical note and image we transferred were all encoded into binary patterns first. In this sense our statement is, indeed, trivial. However, the distribution of the spots denoting the letter and position in the binary pattern is quite different for the various formats. We herein show that different lengths of binary messages which represent text, musical notes or images can be transferred. We have thus changed the "generality" to "versatility" in the abstract. Further, e.g., the depiction of the maps in Fig. 3, is a non-binary format. We have also modified the description in the main text (Lines 43-46, Page 2).

- Lines 42-44: Wrong statement. Quantum computers will not be able to help with brute force attack. They will help with cracking the secure key exchange by solving related computationally hard math problems, such as integer factorization. Then the cyphertext can immediately be cracked without brute force.

Reply: We thank the reviewer for this professional opinion. We have modified the description in the main text and added a corresponding reference (Lines 7-8, Page 2; ref 9, Page 9).

- Lines 44-48: Information theoretically secure quantum and classical physical key exchangers like QKD, KLJN, or just sharing a USB stick containing the key, may indeed have problem with the accessibility part of CIA. However so is the scheme described in the present paper therefore this work is not helping that issue.

Reply: We agree with the reviewer that the accessibility part of CIA is not fully embodied in this work. We introduced DNA linkers that act as password to achieve access control of the message (Figure 3). In our opinion, this addresses the accessibility part of CIA. To clarify that, we modified the corresponding description in the manuscript (Lines 31-32, Page 2).

- Lines 85-89 and 90-109: The manuscript does not clarify the basic elements of secure communication even though it talks about "key" many times. Please exactly specify what is the cypher and what is the shared secret (key) between Alice and Bob and that how is that key shared.

Reply: We thank the reviewer for the suggestion. In the outer-layer encryption (so-called pattern encryption), the key is the permutation of the spot representing the binary digits of the letters or their positions. The cipher is the spot pattern. The spot pattern is the plaintext in the inner-layer encryption (so-called DNA origami encryption, DOE). The key in DOE is the sequence, length and folding of the long scaffold strand. The cipher is the floppy scaffold carrying specific M-strands. The sequence and length of the scaffold can be almost acknowledged through sophisticated sequencing. Therefore, the key size of DOE is calculated based on the folding of the scaffold strand. Both the inner and outer layer encryption are symmetric and the keys are pre-shared in person or by mail. We have added a corresponding description to the main text (Lines 12-13, 20-21, Page 3).

The process can be further illustrated as follows:

This figure has been added to the Supplementary Information (Supplementary Figure 2, Page 10).

- Section "Text communication": The same here. Please identify clearly which elements of the multi-step encoding process are using a shared secret (key) and which part are "built-in" features. Note: in accordance with Kerckhoffs's principle (Shannon's maxim) of secure communicators, only the spontaneously generated key is secure, the rest of the system is known by the eavesdropper.

Reply: As the figure above shows, the first step using a shared key is the encryption of binary numbers into spot patterns in the outer-layer encryption. The key here is the permutation of the spot representing the binary digits of the letters or their positions. The second step using a shared key is the encryption of spot pattern into a scaffold (a long single-stranded viral DNA) carrying certain biotinylated M-strands. The key here is the sequence, length and folding of the long scaffold strand. Besides, a keypad is used to encode the letters into binary numbers. And the intrinsic addressability of DNA origami allows for streptavidin-binding-based steganography. These can be counted as "built-in" features as the reviewer suggested. According to Kerckhoffs's principle, only the keys are required to be kept secret. We have added the description to the Supplementary Information (Supplementary Figure 2, Page 10).

- Section "Integrity ...": Please clarify how the hashing and the given signature will protect the integrity in the light of the Kerckhoffs's principle described above. The only thing that Eve will not know is the secure key (to be specified).

Reply: We thank the author for this professional suggestion very much. We have reclarified the process in the Supplementary Information (Supplementary Figure 22, Page 30) to make it more comply with Kerckhoffs's principle. Besides, we have modified the corresponding description in the manuscript (Lines 40-41, Page 5).

Initially, the hash value is given to them. In this case, the hash value is "GF GG GE GC GB GD BF BG BE BC BB BD EF EG EE EC EB ED DF DG DE DC DB DD CF CG CE CC CB CD FF FG FE FC FB FD" (in crimson box) or "CC CE CF EC EE EF FC FE FF" (in cyan box). Both of them are defined as correct. There is no fixed order between these double-letter segments. When Bob or Mallory obtained the folded DNA origami, they used their passwords (DNA linkers) for assembly of DNA origami dimers. Consequently, the hash value was generated from the splice of double-letter segments. In this communication, the hash value obtained with Bob's password should be the one in crimson box as expected, while that for Mallory should be the one in cyan box. If the integrity of the message is hampered, for example, someone added misleading biotin patterns to mix with the original one, both Bob and Mallory will perceive that. That is because there are no complementary sticky ends to their DNA linkers in the misleading biotin patterns, which will result in extra unrecognizable DNA origami monomers in AFM imaging. On the contrary, if there is a lack of double-letter segments, some DNA origamis could be missing. It has been known that the pattern with the segment "EF" carried the message. Hence, both the Bob and Mallory can detect their target patterns. However, the pattern obtained by Bob is the correct one, while that found by Mallory is wrong. In the overall process, the only thing that needs to be kept secure is Bob's password.

- An essential feature is not characterized: Bit Error Probability. This can be a killer. Please clarify it.

Reply: We thank the reviewer for this suggestion. We have added corresponding discussion on “Bit Error Probability” to the Supplementary Information (Lines 26-42, Page 7; Lines 1-8, Page 8).

The “Bit Error Probability” may be reflected in this work by the accuracy of the spot pattern at each position in AFM imaging. We have conducted multiple tests with message of different lengths as well as two blind tests (Supplementary Figures 17-20). We found that in all tests the major patterns that took up over 50% of the population were the correct ones, but there also was a minority of wrong patterns. The error ratio was relevant to the number of biotinylated spots in corresponding positions since the efficiency of conjugation between biotin and streptavidin is only ~95% (ref. 15 in the main text). Besides, we found that the spot pattern with fewer position spots (spots that indicate the position of the letter) tended to have a higher error ratio. We think this is because once the position spot in another spot pattern is missing, the spot pattern will be wrongly counted in the statistics of the one with the corresponding position spots.

Hence, the bit error ratio is dependent on the number of spots and their distribution in denoting the letters or the positions and thus is different at each position. For example, from the statistics of the message “19120623”, we estimated the Bit Error Probability of each letter as follows:

1st “1”: 42.4%

2nd “9”: 27.7%

3rd “1”: 34.7%

4th “2”: 32.2%

5th “0”: 39.3%

6th “6”: 20.6%

7th “2”: 32.7%

8th “3”: 25.2%

We anticipate that technological advances and new characterization methods will further reduce the bit error probability, so this will not be a fundamental obstacle.

- Section "Versality": As mentioned above, this is trivial though the non-expert media maybe interested. If the above-required clarifications occupy too much space, this section can be dropped.

Reply: We keep this part since the journal Nature Communications does not have a page limit. Also, we present both binary and non-binary (e.g. maps) formats in this part.

Reviewers' Comments:

Reviewer #3:

Remarks to the Author:

The answers addressed the problems I posed though mostly in the supplemental part that only experts will read.

Anyway, I support publication.

Reviewers' comments:

Reviewer #1 (Remarks to the Author):

The answers addressed the problems I posed though mostly in the supplemental part that only experts will read.

Anyway, I support publication.

Reply: We thank the reviewer for this very positive comment.